# Understanding Aboriginal Models of Selfhood: The National Empowerment Project’s Cultural, Social, and Emotional Wellbeing Program in Western Australia

**DOI:** 10.3390/ijerph19074078

**Published:** 2022-03-29

**Authors:** Pat Dudgeon, Kate L. Derry, Carolyn Mascall, Angela Ryder

**Affiliations:** 1Poche Centre for Indigenous Health, School of Indigenous Studies, The University of Western Australia, Crawley, WA 6009, Australia; pat.dudgeon@uwa.edu.au; 2Relationships Australia WA, West Leederville, WA 6007, Australia; Langford Aboriginal Association, Langford, WA 6147, Australia; carolyn.mascall@relationshipswa.org.au (C.M.); angela.ryder@relationshipswa.org.au (A.R.)

**Keywords:** Indigenous psychology, social and emotional wellbeing, suicide prevention, self-determination, cultural identity, cultural safety, Aboriginal health, program evaluation

## Abstract

Culturally safe and responsive interventions that acknowledge Aboriginal models of selfhood are needed. Such interventions empower Aboriginal peoples and communities by increasing self-determination over individual and community social and emotional wellbeing (SEWB). In response to this need, the National Empowerment Project developed the Cultural, Social, and Emotional Wellbeing Program (CSEWB). The CSEWB aims to strengthen SEWB and cultural identity and subsequently reduce psychological distress in Aboriginal peoples. An Aboriginal Participatory Action Research approach ensured community ownership and engagement. Seven research questions and a culturally modified adaption of the Most Significant Change technique informed a thematic analysis of the evaluation content. Aboriginal adults (*n* = 49; 53% ≥50 years, 66% female, 34% male) from three Western Australian urban communities participated in the program evaluation workshops. Participants reported the benefits of enhanced SEWB and reduced psychological distress. This research reaffirms the need for culturally safe programs that acknowledge social determinants of health and are guided by the SEWB framework. Long-term commitment from the government is needed to support such programs.

## 1. Introduction

Truth-telling about colonisation, community empowerment, and reclaiming cultural identity all play a critical role in strengthening the social and emotional wellbeing (SEWB) of Indigenous peoples [1]. A lack of health and mental health services that address SEWB and social determinants of health contributes to the disproportionately high rates of psychological distress, suicide, and health inequity between Aboriginal and non-Indigenous Australians [2]. After decades of continued mistreatment and policy failure by the Australian government and health system, new approaches were needed. Aboriginal-led and community-based projects are appropriate to respond in a culturally safe way to high levels of distress in Aboriginal and Torres Strait Islander communities [3,4,5]. The National Empowerment Project (NEP) was established in 2012 to facilitate Aboriginal and Torres Strait Islander understandings of holistic health and wellbeing. The outcomes of the NEP community consultations were integrated into a Cultural, Social, and Emotional Wellbeing Program (CSEWB). The SEWB framework that underpins the CSEWB reclaims an Aboriginal discourse, acknowledging the role of colonisation in disrupting the mental health and wellbeing of Aboriginal and Torres Strait Islander peoples [6,7]. The CSEWB is a suicide prevention program for Aboriginal and Torres Strait Islander peoples that: (1) promotes SEWB and reduces distress, and (2) empowers communities to address the social determinants of health and support self-determination [8,9]. The current study examines the impact of the learnings from the NEP CSEWB program in Aboriginal participants residing in three Western Australian communities and provides further validation of the SEWB framework as a culturally appropriate model of selfhood for Aboriginal peoples.

Suicide has been described as the greatest public health challenge of our time. The age-standardised rate of suicide for Aboriginal and Torres Strait Islander peoples in Australia is two to three times the rate of non-Indigenous peoples [10]. Reducing these rates has been a public health priority for over a decade [11,12,13]. Suicide deaths are only a fraction of the widespread psychological distress experienced by Aboriginal and Torres Strait Islander peoples since colonisation [14] and constitute a violation of the fundamental human right to health and wellbeing [15]. It is not appropriate to use mainstream models of suicide risk with Indigenous peoples [1,5]. Although some suicide risk factors are common across populations (i.e., social determinants of health), many more complex and cumulative risk factors are derived from the continued effects of colonisation [16,17,18,19]. The historical, political, cultural, and social determinants of health must be acknowledged to abate the disproportionate suicide rates and broader health inequity between Indigenous and non-Indigenous peoples [7,19]. The National Agreement on Closing the Gap has affirmed that to prevent suicide by Aboriginal and or Torres Strait Islander Peoples, policy and programs must address individual and community social and emotional wellbeing [13]. 

In 2009, the Australian government formally endorsed the United Nations (UN) Declaration of the Rights of Indigenous Peoples [15]. Yet, the colonial orientation of health services continues to shape contemporary Aboriginal disadvantage [20], reaffirming the need for investment in Aboriginal and Torres Strait Islander community-led frameworks and interventions [21]. Protection policies grounded in deficit discourse [22], structural racism in the healthcare sector [20], and the lack of truth-telling of past injustices [23] have exacerbated health inequity and deter help-seeking behaviours in Aboriginal and Torres Strait Islander peoples [24]. Health and mental health professions have been complicit in this process. By imposing individualistic constructions of human behaviour on Aboriginal and Torres Strait Islander peoples, mainstream psychology negates Aboriginal and Torres Strait Islander knowledge systems and constructions of identity and holistic health and wellbeing [25]. This is evident in the paucity of culturally safe and responsive services, research methods, and frameworks being utilised by non-Indigenous practitioners [26] and subsequently high rates of emergency department presentations [24,27]. In 2016, the Australian Psychological Society issued an apology to Aboriginal and Torres Strait Islander peoples for failing to uphold the right to equality and cultural recognition [25]. One of the first reconciliation objectives was to transform the deficit narrative by advocating Aboriginal and Torres Strait Islander psychology and the holistic SEWB framework.

Indigenous and mainstream understandings of health and wellbeing differ in important ways. The National Strategic Framework for Aboriginal and Torres Strait Islander Peoples’ Mental Health and Social and Emotional Wellbeing [6,28] contains nine principles that describe the view of whole-of-life health held by Aboriginal and Torres Strait Islander peoples. The first principle (1) states that ‘health is viewed in a holistic context that encompasses mental health and physical, cultural, and spiritual health. Land is central to wellbeing. Crucially, it must be understood that while the harmony of these inter-relations is disrupted, (Aboriginal and Torres Strait Islander) ill-health will persist.’ Aboriginal and Torres Strait Islander peoples’ selfhood and identity are more collective than the individualistic orientation of the West [29]. Belonging to a family and ‘kinship’ group (7) and connection to Country is central to Aboriginal identity and spirituality [30,31,32]. The relational construct of community distinguishes Aboriginal and Torres Strait Islander peoples from other Australians, and the effective functioning of and harmony within the community plays a critical role in Aboriginal and Torres Strait Islander’s wellbeing [30,31,32,33]. Aboriginal and Torres Strait Islander peoples have a deep understanding of these ecological and holistic relationships (9). As a result of these understandings, severe mental ill-health was rare in traditional culture [29]. Yet, since colonisation, the disruption to Aboriginal and Torres Strait Islander’s self-determination and wellbeing has led to unprecedented, intergenerational, and cumulative trauma and loss (4) [16,17,18], which are compounded by the experience of ongoing stressors, including racial trauma, stigma, environmental adversity, and socio-economic disadvantage (6) [19,20]. To be effective, it is essential that Aboriginal and Torres Strait Islander health services are characterised by culturally valid and holistic understandings of wellbeing (3) [1], acknowledge Aboriginal and Torres Strait Islander peoples’ diversity (8), and uphold human rights (5), in particular the right to self-determination (2) [15,34].

In response to suicide rates in the Kimberely, Bardi woman Professor Pat Dudgeon led a community consultation that culminated in the landmark Hear Our Voices report as a part of the Kimberley Empowerment Project [3]. In 2012, the National Empowerment Project (NEP) was established to develop a national Aboriginal-led solution to individual and community distress [4]. The NEP facilitated community empowerment and self-determination by utilising a strengths-based Aboriginal Participatory Action Research approach (APAR) [35]. The NEP engaged 457 community members and leaders to identify ways of addressing the underlying causes of suicide risk by increasing cultural, social, and emotional wellbeing and resilience among Aboriginal and Torres Strait Islander peoples, families, and communities. To this end, a corresponding program was developed through consultation and workshops with 11 Aboriginal communities across Australia. The consultations and workshops recapitulated existing research on Aboriginal and Torres Strait Islander SEWB and determinants of health [36]. During the workshops, it became apparent that participants across all sites strongly felt that explicit inclusion of the word ‘culture’ is necessary to signify the unique world views and values of Aboriginal and Torres Strait Islander peoples and the cultural components that were an integral part of the program. Although SEWB is accepted as the terminology and framework for describing Aboriginal and Torres Strait Islander mental health and wellbeing [7,13], the program was titled the Cultural, Social, and Emotional Wellbeing Program (CSEWB). 

The SEWB model [7] in Figure 1 informed the development of the CSEWB. This model was developed by Aboriginal psychologists and implements a decolonising strategy and perspective that is broader and deeper than the dominant discourse on mental health. It utilises a strengths-based discourse and whole-of-life understanding of health that emphasises wellness, harmony, and balance rather than individual ill-health and symptom reduction [22]. The idea of the self at the centre of the diagram is a collective self. SEWB depicts wellbeing as a series of interdependent relationships between body and mind, family/kinship, community, culture, land/Country, and spirituality/ancestors. Importantly, the model also recognises the concurrent and cumulative influence of social, political, cultural, and historical determinants of health [7]. Cultural identity and collective selfhood are paramount to healing, health, and wellbeing. This model of SEWB was endorsed by the communities involved in the NEP and is recognised as best practice in the SEWB framework [6].

The CSEWB program aims to empower Aboriginal and Torres Strait Islander peoples and strengthen family and community connections and cultural identity. Empowerment is achieved by educating program participants on SEWB and teaching life skills to support positive change (i.e., problem-solving, conflict resolution, communication, goal-setting). Connection to community groups, mentorship, and leadership are also emphasised to strengthen relational networks. Throughout the program, participants are reconnected with family, community, history, and culture using yarning and story-telling techniques. There is also education on positive parenting, family structures, and local and national Aboriginal and Torres Strait Islander history. 

The first CSEWB programs were delivered and evaluated between 2014 and 2016 in Kuranda and Cherbourg, two Queensland communities that participated in the NEP consultations [37]. To ensure cultural safety, APAR was utilised to safeguard the community ownership and the collaborative nature of the program. The Most Significant Change technique [38] was employed to monitor and record complex social change resulting from the program. The use of stories enabled a cultural adaption of the technique that reflects Aboriginal ways of knowing, being, and doing. Thematic analyses of the outcomes of the stories of the Most Significant Change (SMSC) technique revealed strong similarities between the two communities, with eight key themes present overall. Four of the themes related directly to individual wellbeing. These included strengthening: (1) personal strengths, (2) relationships, (3) SEWB, and (4) cultural reconnection. The other four themes related to strategies that address the key issues affecting community wellbeing; these included the need to support: (5) health, (6) family, (7) life skills, and (8) education [37]. The findings provided further validation and support for the SEWB model. 

In addition to SMSC, open-ended interviews informed six case studies. These case studies provided evidence of lived experiences of positive change in the program participants’ aspirations and abilities, as well as increased motivation to overcome the significant challenges and disadvantages endured over their lives. The case studies demonstrated that by strengthening cultural identity through a framework of SEWB, participants experienced an increased sense of belonging and connection to Country and a renewed focus on family and community connections. The key aim of the program was to reduce experiences of psychological distress and hopelessness that may contribute to suicide behaviours and to facilitate a sense of empowerment and agency for participants. Participants’ perceptions of distress (measured through SMSC) were reduced, despite unexpected events that contributed to widespread community grief and loss during the program delivery. 

The CSEWB program had a strong, positive impact on the Queensland communities. However, more research is needed to (1) investigate the efficacy of the CSEWB program in diverse Aboriginal communities across Australia, (2) investigate the underlying structure of Aboriginal and Torres Strait Islander wellbeing, and (3) explore culturally appropriate techniques to measure change in relation to Aboriginal SEWB, including both qualitative and quantitative measures. The current study aims to further validate and develop understandings of SEWB and to measure the impact of the CSEWB. Participants from three communities in Western Australia (WA) were engaged in post-program evaluation workshops. It was hypothesised that participation in the CSEWB program improves participant SEWB, reducing psychological distress and improving agency. It was also hypothesised that thematic analysis of the workshop content would provide further insight into the underlying structure of SEWB and potential mechanisms of change.

## 2. Materials and Methods

### 2.1. Aboriginal Participatory Action Research

Consistent with best practices in research with Aboriginal peoples, APAR was utilised in this study. APAR is a culturally safe and strengths-based research method. By implementing the right to self-determination in research, APAR empowers Aboriginal peoples to describe and share their own reality and to have community ownership of research processes and outcomes [8,35]. Community ownership of knowledge ensures that research outcomes are meaningful and culturally relevant to both Aboriginal communities and researchers. Critically, APAR facilitates genuine partnership and collaboration between researchers and research participants. This requires the employment of Aboriginal community researchers and people who are culturally responsive and able to work in multi-cultural and multi-disciplinary spaces. This decolonising approach to research represents a shift towards a more empowering approach that supports an Aboriginal psychology. This approach also acknowledges gaps in past research, which largely ignored the complex and multifaceted determinants that impact the health and wellbeing outcomes of Aboriginal peoples and communities.

Three Aboriginal community centres were engaged as partners to host the workshops. The community centres and associated regions were (1) Langford Aboriginal Association (Langford, Kelmscott, Gosnells); (2) Darius Wells Centre (Kwinana, Rockingham); and (3) Miya Kaadadjiny (Girrawheen, Balga, Koondoola). These three centres are spread across the Perth metropolitan region (east, south, and north, respectively) to maximise engagement and access. Each region has a community reference group (CRG) and recruited co-facilitators from the local area who had either participated in the program or were respected people within that community. Endorsement and recruitment were facilitated through the CRG membership and Aboriginal community-controlled organisations in the local area. The evaluation workshop facilitators were experienced co-facilitators from the program delivery. Author 4 was one of the community researchers during the initial NEP and has continued this role in managing and facilitating the delivery of the CSEWB program in Perth in partnership with Author 1 at the University of Western Australia.

### 2.2. Participants

Participants who had completed the CSEWB program in Perth were the focus of this research. These participants were recruited to the CSEWB program from the NEP or were referred to the CRG by local service providers. Following the CSEWB graduations, the 82 program graduates were invited to attend the evaluation workshop. In total, 49 adults (70% female, 30% male; 14% were aged between 18 and 25 years of age, 8% 26–35 years; 25% 36–50 years; 53% 50+ years) participated in 1 of 6 workshops (*n* = 8 Kwinana 2019; *n* = 5 Girrawheen 2019; *n* = 9 Langford 2019; *n* = 10 Kwinana 2020; *n* = 8 Langford 2020; *n* = 9 Langford 2021). All workshop participants identified as Aboriginal peoples. None of the program participants in this research identified as Torres Strait Islander. Written informed consent for the use of de-identified data for research and evaluation purposes was obtained from all participants prior to the commencement of each workshop. 

### 2.3. Measures

#### 2.3.1. Semi-Structured Interviews (SSI)

The SSI involved six questions that were co-designed by the CRG to ensure the design and scope was appropriate to both the study aims and communities’ interests. The questions were primarily designed to investigate the participants’ understanding and perception of SEWB and the congruency of the CSEWB program with the SEWB model. The questions were answered by the workshop group together to be consistent with a collectivist orientation. One facilitator and one scribe were present. In addition to the six questions, other probing questions were used to explore the responses and avoid misinterpretation (e.g., ‘In what way do you mean? Can you give an example?’). The questions are shown below.

What does social and emotional wellbeing mean to you?What increases your social and emotional wellbeing?How do you gain a greater understanding of social and emotional wellbeing?How do you get others to understand and strengthen social and emotional wellbeing?How do we engage Aboriginal youth in understanding social and emotional wellbeing?What contributes to Aboriginal people’s social and emotional wellbeing?

#### 2.3.2. Stories of Most Significant Change (SMSC)

Davies and Dart’s [38] Most Significant Change extraction technique was initially developed to monitor and evaluate change processes in organisational settings. This technique is particularly valuable when assessing programs striving for social change and where a focus on learning is desirable [38]. The CSEWB program aims to enhance people’s understanding of themselves, and it was critical that this could be expressed freely and not constructed by the evaluators. To better align with Aboriginal PAR, the original technique was adapted to measure the impact of the intervention using participants’ stories [37]. SMSC is, therefore, a culturally safe, qualitative approach that enables researchers to investigate indicators and processes of change in the lives of the beneficiaries, as well as potential unintended consequences of program participation. SMSC was used in this study to determine if and how participating in the CSEWB program impacted people’s individual understanding and development of their wellbeing. There is not yet a validated and endorsed quantitative measure of the SEWB construct [39,40].

#### 2.3.3. Post-Program Evaluation Form

A post-program evaluation form asking about the purpose and perceptions of the workshop and requesting details for changes to improve program content was collected to measure participants’ satisfaction with the service delivery and content.

### 2.4. Procedure

The CSEWB program ran for approximately 6.5 h one day per week for 12 weeks and concluded with a 1-day evaluation workshop. Complimentary childcare facilities, meals, and transport were provided for participants during program hours in order to maximise engagement and minimise dropout. The program followed the structure outlined in the NEP CSEWB Facilitators Guide. Each group was facilitated by two co-facilitators who were local community members and undertook facilitation training prior to workshop commencement. If participants did not attend, they were able to catch up on the work in their own time or attend a one-on-one session with a facilitator. Culturally responsive counsellors were available to contact during or outside program hours. A community project, determined by participants, was undertaken at the end of the program and before the evaluation workshop. This paper reports on the evaluation workshop following completion of the program. Participants who completed all program content were graduates. 

## 3. Results

### 3.1. Thematic Analysis

A thematic analysis [41] was used to explore the workshop data. Thematic analysis is a method of identifying, examining, and reporting prominent patterns or themes in a dataset. Rather than a passive qualitative process in which themes are thought to emerge independently of researchers’ positioning, thematic analysis is an active qualitative process that enables researchers to actively organise and describe data. To ensure APAR, both facilitators and participants verified the meaning of the information collected during the evaluation workshop. 

In this study, multiple methods were used to enhance the reliability and validity of the thematic analysis. First, two research teams (consisting of Aboriginal and non-Indigenous researchers) conducted an independent thematic analysis of the workshop data. Consistent with [41], the analysis began with a pre-reading of the scribed data to obtain a general sense of the breadth and depth of the content. An initial list of ideas was then compiled and segmented into units of meaning. These were then sorted into potential themes, with some units being merged and some combined to form broad overarching themes. The themes were then reviewed and refined into a coherent pattern of responses, and a ‘thematic map’ was developed. The two research teams then collaboratively discussed any discrepancies in the two thematic constructions, and the final themes were defined and named. To ensure depth and breadth of the emergent themes, the number of references to each theme and the representation between workshops was documented [42]. Then, the thematic construction was reviewed for consistency and comprehensiveness by the research team, adding to the methodological and interpretive rigour of the study [43].

Analysis of the responses from the semi-structured interviews and SMSC showed the emergence of six key themes and fifteen subthemes that characterise SEWB. Each theme and associated subthemes are described in turn below. Figure 2 presents a thematic map of the themes and subthemes in the semistructured interviews. Figure 3 presents a thematic map of the themes and subthemes in the SMSC. The relative strength of each theme and subtheme in the interviews is shown by the size of each theme and sub-themes respective area in each of the Figures. Overall, the themes indicate a strengths-based understanding of SEWB. The distinction between the themes is much less resolute than pictured below, as many constructs are relevant to more than one domain (e.g., camping trips are linked to Country and community; Elders are linked to culture and kinship). Participants’ descriptions of SEWB demonstrated that themes were both interrelated and interdependent.

#### 3.1.1. Theme 1: Culture

The importance of culture in strengthening the understanding and experience of SEWB was the most prominent theme in the participants’ interviews and one of the strongest themes in the SMSC. Participants described that participation in the program gave them an understanding of cultural identity and its role in strengthening SEWB. 

Culture was separable into four subthemes: (a) cultural knowledge, (b) cultural practices, (c) connection to Country, and d) spirituality. Cultural knowledge included a commitment to learning about culture (e.g., ‘know culture’; ‘do workshops’; ‘being a part of Aboriginal groups, and organisations’; ‘spending time with Elders’; ‘respecting Elders’) as well as participating in cultural traditions and cultural forms of healing (e.g., ‘ceremonies’; ‘healers’; ‘lore’; ‘language’; ‘songlines’; ‘dances’; ‘fire; ‘hunting and gathering’; ‘traditional/bush food’; ‘wildflower treatment’). The importance of ‘yarning’ or ‘story-telling’ was emphasised and linked to respecting Elders and connecting to culture, community, and family. 

Spending time in the bush or on traditional homelands was important. ‘Shoes off, on boodja (Country)’, ‘going out bush with Elders’, ‘going back to Country’, ‘gatherings on Country’, and ‘acknowledging Country’ facilitated (re)connecting with Aboriginal culture through land and place. Participants shared a sense of empowerment in embracing cultural practices and connection to Country that could be expressed through activities such as Welcome to Country ceremonies.

Connecting to spirituality was generally described indirectly, as ‘what I believe’, or through mindful practices (e.g., ‘meditating’; ‘reflecting’; ‘sitting at sunset’; ‘watching’), or ‘time out’ (e.g., ‘me time, alone time’; balance’; ‘getting away’). These quiet practices were described as ‘relaxing’, ‘spiritual’, or ‘connecting to spirit’ when questioned how they are associated with enhanced wellbeing (e.g., ‘I’ve been listening to more healing music to keep calm’). 

SMSC described that as a result of remembering history, a new appreciation of culture and how it impacts Aboriginal wellbeing was found (e.g., ‘It has taught me that yes, we have a culture and that our culture is very strong and is still alive and it lives still within us’). This included ‘recognising,’ ‘respecting,’ and ‘trusting Elders and listening to their stories’. In the words of one individual, ‘I have learnt a great deal about how important it is to carry on our culture, traditions and language to our younger generations and not let it disappear’. This new knowledge and responsibility bolstered self-esteem, ‘I have more insight into my cultural learnings about cultural stories and more confidence into every aspect of my life’. This seemed to some to be something that was not consciously expressed prior to the program (e.g., ‘they taught me more about our culture and traditions in 12 weeks than I ever knew in my lifetime’; ‘knowledge is needed–knowing the history to pass on’; ‘I am now comfortable in teaching my nieces/younger mob our ways’). Even if connection to kinship had been thwarted due to Stolen Generations membership and forced child removals, wellbeing could be nourished through a sense of attachment to Country, as described by one participant, ‘while Country still accepted me, I was forever rejected by those I was meant to belong to, continually crushing my spirit, learning that Country accepts unconditionally’. Similarly, there was an appreciation of ancestors as an important form of attachment and comfort, ‘Family inspiration through time’; ‘acknowledgement through your old ones.’

#### 3.1.2. Theme 2: Community

Community was also a prominent theme in the interviews. Participants described that completing the program significantly contributed to their interest in and perceived benefits of participating in and engaging with their communities. For many participants, sharing and promoting information about the program was central to building community support. The opportunity to participate in a community project reinforced a sense of connectedness and belonging to others, asserting themselves as a proud member of their community. The sense of belonging and connection to community was clearly linked to an increased sense of wellbeing.

Community involved two subthemes: (a) seeking support from others and (b) giving support to others. Seeking support involves being actively engaged in the community as well as looking for and accessing support services (e.g., ‘flyers’; ‘websites’; ‘reading’; ‘find out’). Participants described an increased awareness of culturally appropriate services where they felt comfortable. This resulted in stronger community connection and confidence in reaching out. Communication was integral to this process (e.g., ‘asking for help’; ‘talk about it’; ‘know each other’; ‘listening’; ‘ask questions’; ‘share our knowledge’), as was active participation (e.g., ‘be actively engaged’; ‘stay connected’; ‘be social’; ‘reduce isolation’). Social and sporting activities (e.g., ‘men’s and women’s groups’; ‘social groups’; ‘darts’; ‘camps’; ‘community projects’) allowed participants to feel connected and an increased sense of belonging. Giving support was even more important, as participants felt a new responsibility to ‘encourage’ and ‘motivate’ others (e.g., ‘check in on people’; ‘catch a bus together’; ‘guidance’; ‘mentor’; ‘including’; ‘offer support’; ‘ask are you okay?), and be a positive role model within their communities (e.g., ‘to share our knowledge and our strength’; ‘teaching the next generation’; ‘build community strength’; ‘to give back’; ‘volunteer’; ‘perform and teach in schools’).

In the SMSC, many participants described that participating in the community, rather than being isolated, was a significant change for them (e.g., ‘doing a lot more community work’; ‘build confidence and sharing and caring more’; ‘made myself available to share and spent time with others’; ‘would like to be a strong role model always engaging with people’). The importance of community strength in cultural and personal wellbeing was also described (e.g., ‘I learnt that our communities and people have to stand together to achieve success for our youth’; ‘it has changed my way of thinking and to help other people’; ‘I had an intention of learning about trauma (Aboriginal) but had walked away with more knowledge of my culture and connection and how that feels in the community’; ‘I have enhanced my association of togetherness and will continue to highlight this in my life.’).

#### 3.1.3. Theme 3: Kinship

Participants commonly described the importance of kinship and connectedness with family and friends. Knowing who their mob is and where they come from empowered people to feel more confident in their cultural identity. The intricate knowledge of complex family lines helped to strengthen participants’ sense of belonging and was linked to reduced feelings of isolation and increased wellbeing. 

This theme was separable into two subthemes: (a) family connections and (b) developing positive relationships. Spending time with family was commonly mentioned as integral to wellbeing. Often, this involved increasing the amount of time spent with family (e.g., ‘family barbeques’; ‘family catch ups’; ‘being there for grandparents’; ‘engaging with parents’; ‘family reunions’), but also having a deeper understanding of family (e.g., ‘old photos’; ‘family knowledge’; ‘connecting with family history’; ‘family wellness’, ‘values’; ‘promote healing by talking to our mob). Distinctions were made around the influence of different family members. Participants identified that seeking out positive family members was critical to maintaining a sense of wellbeing. 

Positive relationships included developing positive relationships (e.g., ‘taking time to build relationships’; ‘friends with like-minded people’; ‘saying no to peer pressure’; ‘trust’) and developing interpersonal skills (e.g., ‘even if don’t agree can listen and respect peoples decisions and viewpoint’; ‘being open minded’; ‘acceptance of others’; ‘empathy’; ‘not judgemental’). Positive relationships also included positive parenting practices (e.g., ‘involving kids in positive activity’; ‘don’t just keep telling them they did wrong’; ‘feeling loved and valued’; ‘it’s okay to make mistakes but learn from it’; ‘teach structure and responsibility’; ‘teach healthy relationships’; ‘make them feel they are in a safe space’) and positive intimate relationships ‘healthy sexual relationships’.

The negative impact of unhealthy relationships and a lack of effective interpersonal skills prior to the program was described in SMSC. Participants described ‘internal issues of identity and belonging’ and the negative impact of feeling a thwarted sense of belonging in their communities ‘being rejected by those I was meant to belong to was crushing my spirit’. By increasing interpersonal skills during the program, participants felt more empowered to deal with conflicts (e.g., ‘I have gained knowledge in how to deal with difficult situations’; ‘I’ve been feeling a lot stronger within myself by using tools from these workshops’). Further, participants often described the significant change of learning more about their family history (e.g., ‘I started my family tree ‘cause I didn’t know where to start’; ‘I went against my mum and decided to go to my nan’s home place’). The negative impact of government assimilation and removal policies on family and the motivation to be a better parent were also commonly described (e.g., ‘nearly all are affected by government policies of the past, sometimes for several generations’; ‘separation of families and institutionalisation has led to poor parenting in some, this has had an ongoing effect on their children to grandchildren’; ‘because I know my sons need me to be the best person I can be to be able to raise them right’; ‘more confident as a mum and a role model for my children’; ‘this program, has helped me, my children and grandchildren and family are always there’).

#### 3.1.4. Theme 4: History

Learning about and accepting Aboriginal history and the continued impact of social determinants was a strong theme in participants SMSC. History was separable into two subthemes: (a) accepting the past and (b) understanding the present. 

Being able to understand one’s history (e.g., ‘historical determinants’; ‘the Stolen Generations’; ‘government policies’; ‘land rights’; ‘the right history’) and current social determinants (e.g., ‘house and roof’; ‘employment’; ‘finances’; ‘education’; ‘jail’; ‘negative impact of drugs and alcohol’; ‘dealing with racism’) were not necessarily considered to be core dimensions of the wellbeing construct, rather they were described as a catalyst for change and a necessary component to ‘accept’, ‘heal’, and ‘move forward’. Participants commonly described having a prior awareness of historical context (e.g., ‘I thought I knew a lot’), yet the program was able to ‘put it all together’ and show ‘the connections’ in regard to ‘how the past effects the future’ and identify reasons for contemporary disadvantage (e.g., ‘everyone has awareness but now I can explain the connection’; ‘by connecting it all it makes life better’; ‘awareness of what is going on’; ‘understand where prejudice came from’; ‘overcoming shame’; ‘Closing the Gap’). 

The SMSC described that the program allowed them to ‘remember the past’ and ‘overcome the fear of asking family questions about the Stolen Generation’ and ‘not be shame’. The stories commonly described the necessity of ‘looking back to move forward’ and ‘learning from the past to strive for tomorrow’. Understanding the impact of the past in their present lives allowed for a greater understanding of self (e.g., ‘it really affects me and now I know why’; ‘As a young child I was taken away and sent to a mission… I lost my culture’; ‘doing this course helped me understand more about myself than I knew’), tolerance and empathy towards others (e.g., ‘learned not to judge harshly’; ‘knowing the background of the story put a different view of the overall story’), and a renewed determination towards a better future (e.g., ‘learning how to stop intergenerational trauma’; ‘learning that no matter how much racism there is in our country, I am and will always be proud’; ‘I have always been aware of the difficulties Aboriginal people have faced and will always face but I have learnt that there are strong individuals and communities still fighting for our rights to be heard’). 

#### 3.1.5. Theme 5: Health

The importance of maintaining one’s health was another key theme, although relatively minor in strength. Many participants highlighted how completing the course equipped them with insight into and education around their health. The program provided a holistic perspective on health; thus, connection to the other themes was critical to maintaining good health and wellbeing. Participants also expressed that the program helped them to identify with healthier lifestyles.

The health theme was separable into two subthemes: a) healthy body and behaviours and b) healthy mind and emotions. Healthy body and behaviours involved being active (e.g., ‘sport’; ‘exercise’; ‘going for walks’; ‘keep out of trouble’), healthy lifestyles (e.g., ‘drinking water’; ‘diet’; ‘good food’; ‘no alcohol or drugs’), and accessing healthcare providers (e.g., ‘medical check-ups’; ‘taking medications’). Mental or emotional wellbeing included mental health (e.g., ‘counselling’; ‘attending health workshops’; ‘Mental Health First Aid’; ‘accessing supports’; ‘less worry and anxiety’; ‘stress’; ‘feelings’; ‘controlling emotions’; ‘understand emotions’) and positive lifestyle practices (‘getting enough sleep’; ‘think carefully’; ‘express yourself’; ‘art–putting it on paper’). 

SMSC revealed that participants often experienced poor mental health prior to the program (e.g., ‘I have been in a dark place for a long time’; ‘my negative thought processes were detrimental to me being able to maintain a healthy lifestyle as I struggled to escape from my own psychological distress’). Participants often described positive ways that personal health practices changed during the program and demonstrated an increased sense of control over their health and wellbeing (e.g., ‘my health is getting a lot better’; ‘look after myself, physically, mentally and emotionally’; ‘I have slowed down and taken good care of myself.’; ‘spending quiet time in my garden and have gone back to painting with oils’; ‘I didn’t fully understand procrastination and how it affected my life. Putting things off was a big one for me that I do a lot, I have started to overcome that now.’) and how personal health became of greater value (e.g., ‘healthy mind, body, spirit needed to remain strong therefore better able to look after family and help in the wider community’). 

#### 3.1.6. Theme 6: Self

Self was a strong theme in participants SMSC, but was a weaker theme in the interviews. Participants commonly described a positive change in their sense of self and described that the program had renewed their interest in living a happy life and achieving self-development. Completing the program provided many participants with strategies to increase their daily happiness and achieve their longer-term goals. 

This theme was separable into three subthemes: (a) self-confidence, (b) experiencing joy, and (c) personal development. The increase in self-esteem was emphasised by participants, most often described as increased ‘confidence’, and similarly as ‘empowerment’, ‘pride’, and a ‘sense of identity’ and ‘feeling complete’ or ‘whole’. In addition, participants described a renewed sense of joy; this could be emotionally through ‘being happy’, ‘laughter’, and ‘positivity’, and enjoying hobbies (e.g., ‘music’; ‘singing and dancing’; ‘shopping’; ‘grow a garden’; ‘passion–do something because you love doing it). 

The theme of self was less prominent in the interviews than in SMSC. Participants commonly described becoming a changed person through the course (e.g., ‘I’m back to the old me’; ‘I had lost the true me’; ‘to come back to my old self’; ‘my healing’; ‘taking good care of myself’), in that they developed a sense of self-love, empowerment, and pride, and an improved capacity for self-care. Although many described the sense of ‘a new beginning’, this did not necessarily imply smooth sailing ahead. Indeed, many also acknowledged circumstances of hardship that would persist, yet expressed optimism in facing challenges (e.g., ‘It’s a long, long climb but I am going to get there’; ‘I’ve got the will to live and the fight for life’; ‘it has given me back the desire to have a life–I haven’t wanted one for almost all of my 65 years’; ‘Strength for good things and not so good things’, ‘for the first time ever I felt a sense of hope for my future’). The new confidence was often described as ‘finding a voice’ and being able to speak out, speak up, or stand up. ‘Being strong in myself’ was seen as necessary to lead others and teach others and fulfil responsibilities to family; in the words of one participant, ‘I constantly left myself to last and it wasn’t until it was too late that I realised that if I don’t look after myself first I can’t look after my own family’. 

Intertwined with this new confidence was a determination to overcome the negative impact of social determinants through personal development. Personal development involved a more holistic view of relational success (vs. Western ideas of success), education (both formal and informal learning), employment (both paid and volunteering), and incorporated other examples of motivation and life skills (e.g., ‘completing/submitting a tax return’; ‘getting a license’). Willingness to engage in future training demonstrates the value participants put on self-development and life skills and their confidence in pursuing their goals (e.g., ‘Now I think it’s better to give things a go or give it a try no matter the outcome’; ‘After last year volunteering in little peeps [peoples] ball and completed Cert II Business, I’m now studying Cert III in Community Services and volunteering at [community service]’; ‘The course helped me get a job which I still have’; ‘Never had a drivers’ licence for decades (until now)’). The program played a critical role in facilitating these changes. 

A feedback form was also administered to participants at the end of the workshop. The content of the evaluation feedback is reported in Table 1.

## 4. Discussion

Novel approaches to suicide prevention are needed in Aboriginal and Torres Strait Islander communities. To be effective, these approaches must work within a decolonial SEWB framework [13]. This project used an APAR approach to investigate the efficacy of the CSEWB program in increasing participants’ reports of empowerment and SEWB, as well as reducing perceptions of psychological distress. Empowerment was examined using a thematic analysis of the participant’s stories of most significant change. 

In addition to participant empowerment, the SEWB construct was explored using a thematic analysis of six interview questions answered collectively by the workshop groups. Using thematic analysis, 6 themes and 15 subthemes emerged that describe SEWB. The six overarching themes (community, culture, kinship, history, health, and self) reflect the domains and determinants described in the SEWB model [7]. The 15 subthemes 1) community (giving and seeking support); 2) culture (connection to spirituality, to Country, cultural knowledge, and cultural practices); 3) kinship (family connections and developing positive relationships); 4) history (understanding/healing history and determinants of health); 5) health (body/behaviour and mind/emotions), and 6) self (self-esteem, experience joy, personal development) provide further delineation of these core components of Aboriginal wellbeing. These domains of SEWB are comparable to other models of Aboriginal wellbeing (e.g., [30,31,44]).

The CSEWB program is a successful, Aboriginal-led community initiative that builds on the strong community-based foundation of the NEP [3,4,23]. The program aims to not only strengthen the SEWB of program participants but also extend to family and community wellbeing as a result of individual empowerment in addition to the benefits of APAR methodologies that ensure community partnership and governance [35]. This research project confirmed the diversity and consistency of positive impacts that program graduates have experienced and developed further in the time following the completion of the program. Graduates described significant change over the course of the 12-week program in the six thematic domains of wellbeing. These positive impacts demonstrate the capacity to successfully address a variety of issues that participants, their families, and their communities are experiencing. The positive impacts included education about and understanding of the ramifications of national, local, and personal histories and increased awareness of the complex interactions between risk and protective factors of wellbeing and the importance of cultural reclamation and holistic understandings of health and healing. This was reflected in the emerging themes taken from the group workshop and the SMSC. The SMSC demonstrated substantial healing through enhanced empowerment, strengthened SEWB, and reduced psychological distress. This indicates that the CSEWB program may have transformational and sustained benefits for Aboriginal peoples. 

Future research should aim to develop and validate culturally appropriate scales that are governed by and developed using appropriate research methodologies and directly contribute to an Aboriginal understanding of SEWB. For example, these scales could ask, over the last four weeks, how much had they had connected with culture, connected with spirituality, sought or provided community support, or felt kinship in positive relationships. Community-based APAR research from Aboriginal-led research teams partnering with the community would be integral in understanding the best ways to frame these questions, the mode of delivery for such questions, and a meaningful scale on which to assess these factors [35].

This research must also be viewed in light of its limitations. This paper reports on a sample of 49 interviews and SMSC. This sample provides considerable detail about these individuals’ experiences. From this information, details on the relative importance of the emerging themes to an Aboriginal sense of wellbeing was provided. Larger samples could provide further detail on the strength of relative importance of each theme to wellbeing. Further, longitudinal data collected on these themes and the timing of related factors, such as traumatic life events, stress, and use of coping skills, could elucidate the trajectory of these needs.

## 5. Conclusions

This research reaffirms the utility of culturally appropriate programs that acknowledge social, historical, political, and cultural determinants of mental health in Aboriginal peoples and supports the utility of the SEWB framework and APAR methods to guide the development of these programs and assess their efficacy. Beyond the Australian context, this research also aligns with the UN 2030 Sustainable Development Goal 3: good health and wellbeing and the UN Declaration on the Rights of Indigenous Peoples (2007). Future research would be well placed to develop an instrument or method for assessing SEWB domains and determinants of health from an Aboriginal lens, incorporating themes from the CSEWB program evaluation alongside the SEWB model [7] and National Strategic Framework for Aboriginal and Torres Strait Islander Peoples’ Mental Health and Social and Emotional Wellbeing [6]. Social and emotional wellbeing has been explicitly included as a target in the National Agreement on Closing the Gap and identified as the best strategy towards suicide prevention [13], yet the National Strategic Framework for Aboriginal and Torres Strait Islander Peoples’ Mental Health and Social and Emotional Wellbeing [6] is yet to be renewed, funded, and implemented. Long-term commitment from the government is needed to work in partnership with Aboriginal and Torres Strait Islander peoples to build the community controlled sector, ensure Aboriginal and Torres Strait Islander self-determination and data sovereignty, and support the development and distribution of such programs and future examinations of factors that may impact the long-term trajectory of wellbeing [13]. 

## 6. Patents

This section is not mandatory but may be added if there are patents resulting from the work reported in this manuscript.

## Figures and Tables

**Figure 1 ijerph-19-04078-f001:**
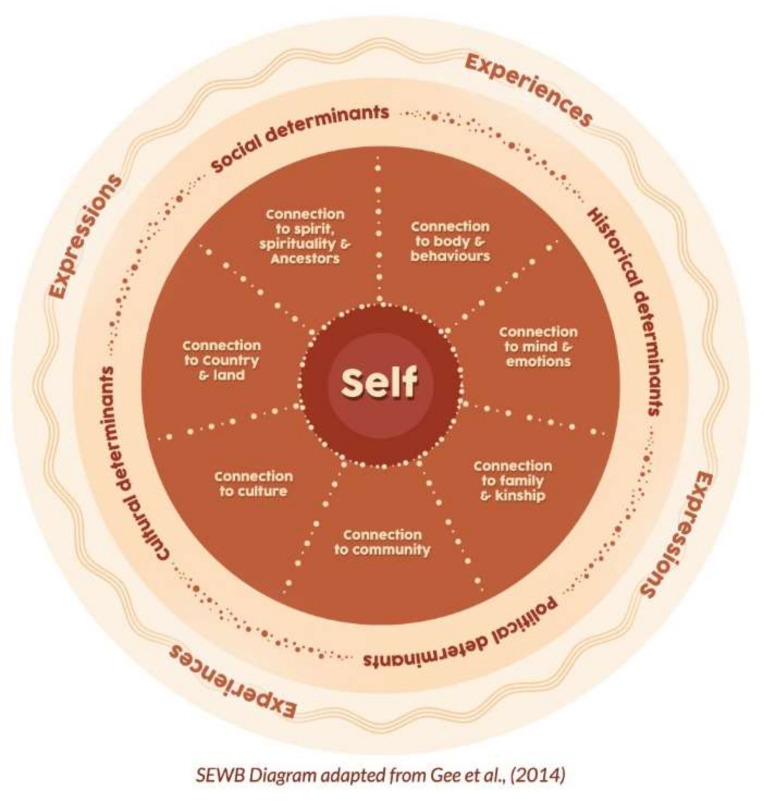
A visual representation of social and emotional wellbeing [7].

**Figure 2 ijerph-19-04078-f002:**
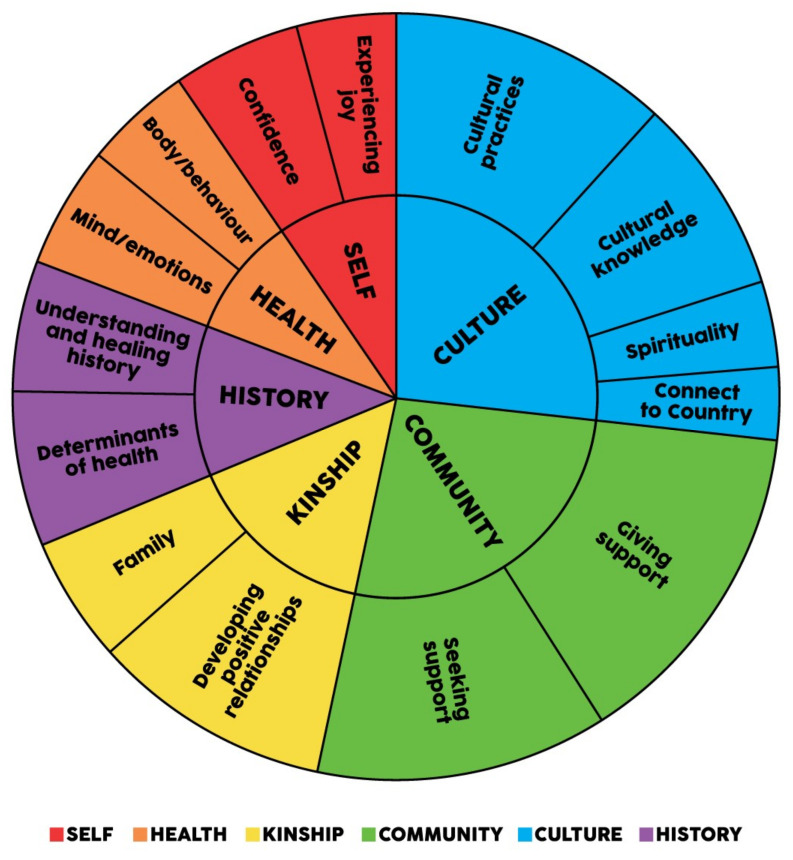
Thematic map of semi-structured interviews.

**Figure 3 ijerph-19-04078-f003:**
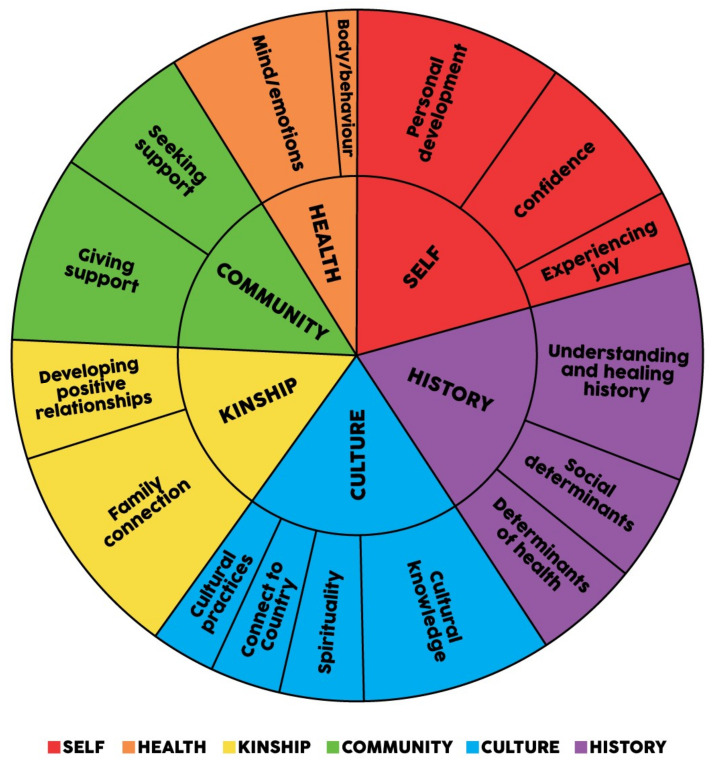
Thematic map of SMSC.

**Table 1 ijerph-19-04078-t001:** CSEWB program evaluations.

	Yes	Unsure	No
I would like the workshop to be longer or shorter	5%		95%
The purpose of the workshop was clear to me	83%	17%	
I felt that I was heard and able to have my say	100%		
The facilitator was respectful and inclusive of all participants	95%	5%	
I felt that the workshop was useful	95%	5%	
After participating in this workshop:			
I have a better understanding of Cultural, Social and Emotional Wellbeing	93%	7%	
I know more about the services that deliver CSEWB in my community	85%	10%	5%

## Data Availability

Data supporting reported results can be found by contacting the corresponding author.

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
