# Peer review of "Understanding Aboriginal Models of Selfhood: The National Empowerment Project’s Cultural, Social, and Emotional Wellbeing Program in Western Australia"

_ijerph, 2022, doi:10.3390/ijerph19074078_

Round 1

Reviewer 1 Report

This article describes a Cultural, Social and Emotional Well-Being (C/SEWB) programme conducted in Australia for the benefit of Indigenous people. The programme is based on Aboriginal Participatory Action Research (APAR) methodologies. It adopts a decolonising approach (rather than relying on dominant discourses on mental health) and yielded positive results, the article concludes.

While following and explaining a specific sociological and psychological methodology, the article is clearly written so that it is accessible and of interest to non-specialists of the field. This is important to give broader resonance to the work that is undertaken in the programme by reaching a wider readership. Beyond the Australian national context, this kind of work also intersects more largely with United Nations Sustainable Development Goals (SDG 3: good health and well-being) and will be of interest to many.

I have noted only a few typos which I will list here:

350: they was described > were

369: it really effects me > affects

370: more then I knew > than

389: processeswere > add space

423: their confidence is pursuing their goals > in

Author Response

Response to Reviewer 1 Comments

Point 1: This article describes a Cultural, Social and Emotional Well-Being (C/SEWB) programme conducted in Australia for the benefit of Indigenous people. The programme is based on Aboriginal Participatory Action Research (APAR) methodologies. It adopts a decolonising approach (rather than relying on dominant discourses on mental health) and yielded positive results, the article concludes.

While following and explaining a specific sociological and psychological methodology, the article is clearly written so that it is accessible and of interest to non-specialists of the field. This is important to give broader resonance to the work that is undertaken in the programme by reaching a wider readership. Beyond the Australian national context, this kind of work also intersects more largely with United Nations Sustainable Development Goals (SDG 3: good health and well-being) and will be of interest to many.

Response 1: Thank you for your encouraging comments and observations. We have noted the relevance to UN SDG3 in the Conclusions section.

Point 2: I have noted only a few typos which I will list here:

350: they was described > were

369: it really effects me > affects

370: more then I knew > than

389: processeswere > add space

423: their confidence is pursuing their goals > in

Response 2: Thank you for noting these minor typos. We will ensure they are fixed in the manuscript when re-submitting.

Reviewer 2 Report

This research is an important contribution to the development of much needed culturally-appropriate techniques and assessment in the field of Indigenous mental health. It is well structured and explains clearly the research process and outcomes. 

It would be useful to have some background on how the WA based project came about and how researchers came to be involved with the project, their roles and contributions (in Materials and Methods or the previous paragraph introducing the WA project).

The Conclusion states "Long-term commitment from government is needed to support the development and distribution of such programs and future examinations of factors that may impact the long term trajectory of wellbeing. " (lines 497-8)

Do the authors have suggestions re how to reach out to government ?

Specific comments:

Line 194: refers to "seven questions" , but lines 190, 197-202 indicate six questions.

Line 438, Table 1: A key to the abbreviations in the table would be useful.

Author Response

Response to Reviewer 2 Comments

Point 1: This research is an important contribution to the development of much needed culturally-appropriate techniques and assessment in the field of Indigenous mental health. It is well structured and explains clearly the research process and outcomes. 

It would be useful to have some background on how the WA based project came about and how researchers came to be involved with the project, their roles and contributions (in Materials and Methods or the previous paragraph introducing the WA project).

The Conclusion states "Long-term commitment from government is needed to support the development and distribution of such programs and future examinations of factors that may impact the long term trajectory of wellbeing. " (lines 497-8)

Do the authors have suggestions re how to reach out to government ?

Response 1: Thank you for your thoughtful and considered comments.

The WA project commences in response to the high rates of suicide and subsequent community distress in the Kimberely region of WA. “Hear Our Voices” was the first report produced from the Kimberely Empowerment Project which grew to be the National Empowerment Project.

92-94. In response to suicide rates in the Kimberely, Bardi woman Professor Pat Dudgeon led a community consultation that culminated in the landmark Hear Our Voices report as a part of the Kinberely Empowerment Project [3]. In 2012, the National Empowerment Project (NEP) was established to develop a national Aboriginal-led solution to individual and community distress [4].

Author 1 led the research throughout, Author 3 was a research associate at UWA at the time, assisting with the NEP. Author 4 was involved in the initial community consultations at the Perth NEP site and became a key co-researcher during the NEP. Author 2 led the data analysis of the evaluation and writing of the Perth workshops. All authors are still involved in the delivery of the CSEWB program in Perth.

183-184. Authors 4 was one of the co-researchers during the initial NEP and has continued this role in managing and facilitating the delivery of the CSEWB program in Perth in partnership with Author 1 at the University of Western Australia.

The SEWB policy framework described in the introduction has been accepted by Government, but has not yet been funded. In 2020, SEWB was included as a key target in the National Agreement on Closing the Gap. SEWB has also been explicitly linked to suicide prevention in this core policy document. This has influenced Government to renew an Aboriginal-specific suicide prevention plan based on Aboriginal research and evidence. Policy documents like these are critical to funding being allocated to Aboriginal community-controlled organisations, and funding support being allocated to Aboriginal-led research, Aboriginal data sovereigty, and community partnered programs.

500-515. 5. Conclusions

This research reaffirms the utility of culturally-appropriate programs that acknowledge social, historical, political, and cultural determinants of mental health in Aboriginal peoples and supports the utility of the SEWB framework and APAR methods to guide development of these programs and assess their efficacy. Beyond the Australian context, this research also aligns with the UN 2030 Sustainable Development Goal 3: good health and wellbeing and the UN Declaration on the Rights of Indigenous Peoples (2007). Future research would be well placed to develop an instrument or method for assessing SEWB domains and determinants of health from an Aboriginal lens incorporating themes from the CSEWB program evaluation, alongside the SEWB model [7] and National Strategic Framework for Aboriginal and Torres Strait Islander Peoples’ Mental Health and Social and Emotional Wellbeing [6]. Social and emotional wellbeing has been explicitly included as a target in the National Agreement on Closing the Gap, and identified as the best strategy towards suicide prevention [13], yet the National Strategic Framework for Aboriginal and Torres Strait Islander Peoples’ Mental Health and Social and Emotional Wellbeing [6] is yet to be renewed, funded, and implemented. Long-term commitment from Government is needed to work in partnership with Aboriginal and Torres Strait Islander peoples, to build the community controlled sector, to ensure Aboriginal and Torres Strait Islander self-determination and data sovereignty, and support the development and distribution of such programs and future examinations of factors that may impact the long term trajectory of wellbeing [13].  

Point 2: Specific comments:

Line 194: refers to "seven questions" , but lines 190, 197-202 indicate six questions.

Line 438, Table 1: A key to the abbreviations in the table would be useful.

Response 2: Thank you for noting these typos. We will ensure they are fixed in the manuscript when re-submitting. Rather than adding a key to abbreviations, we spelled them out in the Table.
